# Histological, Transcriptomic, and Functional Analyses Reveal the Role of Gibberellin in Bulbil Development in *Lilium lancifolium*

**DOI:** 10.3390/plants13212965

**Published:** 2024-10-24

**Authors:** Shanshan Du, Mengdi Wang, Jiahui Liang, Wenqiang Pan, Qianzi Sang, Yanfang Ma, Mengzhu Jin, Mingfang Zhang, Xiuhai Zhang, Yunpeng Du

**Affiliations:** 1School of Life Sciences, Jilin University, Changchun 130118, China; duss1322@mails.jlu.edu.cn; 2Institute of Grassland, Flowers and Ecology, Beijing Academy of Agriculture and Forestry Sciences, Beijing 100097, China; mengd_wang@163.com (M.W.); jiahuiliang1230@163.com (J.L.); pwq211@126.com (W.P.); sangqianzi200004@163.com (Q.S.); 18845303208@163.com (Y.M.); jinmengzhu8@126.com (M.J.); zhangmingfang@baafs.net.cn (M.Z.); zhangxiuhai@baafs.net.cn (X.Z.); 3College of Landscape Architecture, Beijing Forestry University, Beijing 100083, China; 4Ornamental & Edible Lily Engineering Research Center of National Forestry and Grassland, Beijing 100097, China; 5Agriculture College, Yanbian University, Yanji 133002, China

**Keywords:** *Lilium lancifolium*, bulbil development, gibberellin, transcriptome analysis, *LlGA20ox2*

## Abstract

Lily bulbils, advantageous axillary organs used for asexual reproduction, have an underexplored developmental mechanism. Gibberellins are known to participate in bulbil development, but the regulatory mechanisms remain unclear. In this study, exogenous gibberellin (GA_3_) significantly increased the bulbil length, width, and weight by raising the endogenous gibberellin levels and elongating the scale cells. Transcriptomic analysis identified *LlGA20ox2*, a key gibberellin biosynthesis gene, which was upregulated during bulbil development and significantly responsive to GA_3_ treatment. Given the similarities in bulbil and bulblet development, we determined the roles of *LlGA20ox2* using a bulblet system. Silencing *LlGA20ox2* in bulblets inhibited development by reducing the cell length, while overexpression increased the bulblet length and width. In the gibberellin signaling pathway, we identified two key genes, *LlGID1C* and *LlCIGR2*. Silencing these genes resulted in phenotypes similar to *LlGA20ox2*, inhibiting bulblet development. Further transcriptomic analysis revealed that gibberellin-responsive genes were enriched in the glucuronate pathway, pentose phosphate pathway and galactose metabolism pathways. Most of these differentially expressed genes responded to gibberellin and were highly expressed in later stages of bulbil development, suggesting their involvement in gibberellin-regulated bulbil growth. In conclusion, we preliminarily explored the mechanisms of gibberellin regulation in bulbil development, offering significant commercial potential for new lily reproductive organs.

## 1. Introduction

*Lilium lancifolium* possesses significant ornamental, edible, and medicinal value. It is cultivated globally, with an annual production value of approximately CNY 6 billion [1,2]. As a natural triploid plant, its main mode of reproduction is asexual. Bulbil propagation, due to its high propagation coefficient, short propagation cycle, and low virus content, has gradually become the primary propagation method for *L*. *lancifolium* [3,4].

Bulbils arise from the axillary meristem (AXM) located at the junction of the leaf and stem, which represents a form of axillary bud (AXB). Their development can be broken down into two primary phases: the initiation of the AXM and the subsequent growth of the bulbil [5,6,7]. The AXM is generated by both periclinal and anticlinal cell divisions of the parenchyma tissues situated beneath the epidermis at the petiole base. As the AXM continues to divide, it differentiates into the bulbil, which contains an apical meristem and scales, while storing starch granules in its mature state [7,8]. Although considerable strides have been made in deciphering the regulatory pathways of bulbil initiation [7,9,10], the mechanisms controlling bulbil growth remain largely unknown.

As aerial bulblets, bulbils undergo a developmental process similar to lily bulblets and stem-borne bulbils, regulated by hormones such as auxins, cytokinins, and gibberellins [2,9,11,12,13]. Among these, the role of gibberellin should not be overlooked. Gibberellins (GAs), plant hormones with a tetracyclic diterpenoid structure, are involved in key developmental processes such as seed germination, stem elongation, leaf expansion, pollen maturation, and flowering induction [14,15]. GA_1_, GA_3_, GA_4_ and GA_7_ are the primary active gibberellins in most plant species [16,17]. *GA20oxs* (GIBBERELLIN 20 OXIDASEs) and *GA3oxs* (GIBBERELLIN 3 OXIDASEs) are key enzymes in the gibberellin biosynthesis pathway, while *GA2oxs* (GIBBERELLIN 2 OXIDASEs) are involved in its metabolism [18,19]. Mutations or knockdown of *GA20oxs* and *GA3oxs*, or overexpression of *GA2oxs*, lead to dwarfism in rice [20,21]. In the gibberellin signaling pathway, GA binds to the soluble receptor GID1 (GIBBERELLIN INSENSITIVE DWARF1), promoting its interaction with the transcriptional repressor DELLA protein, SLR1 (SLENDER RICE 1) [21,22,23]. The F-box protein GID2 (GIBBERELLIN INSENSITIVE DWARF 2) helps form the SCF (SKP-CULLIN-F-BOX) E3 ubiquitin ligase complex, which polyubiquitinates DELLA proteins, leading to their degradation by the proteasome [18,24].

GAs exhibit dual roles in AXB outgrowth and maturation. In pea, Arabidopsis, rice, turfgrass, and hybrid aspen, GAs may negatively regulate AXB outgrowth [21,25,26,27,28]. GA-deficient mutants in Arabidopsis, rice, and pea exhibit higher branching or tillering levels than wild-type plants. Overexpression of GA catabolic genes similarly reduces GA levels, resulting in increased branching or tillering [21,29]. In Arabidopsis, gibberellins repress AXB formation by modulating DELLA-SPL9 (SQUAMOSA-PROMOTER BINDING PROTEIN LIKE 9) complex activity [27]. Mutants with disrupted DELLA proteins, the primary negative regulators of GA signaling, show reduced stem branching and altered branching patterns [30]. In contrast, in *Rosa* sp., strawberries, and *Jatropha curcas*, GAs may act as positive regulators of AXB elongation [31,32,33,34]. For instance, *FveGA20ox4* was identified as a key gene determining whether strawberry AXBs become runners (elongated branches) or branch crowns (short branches) [33]. The role of GA in bulblet and bulbil development also remains controversial. Elevated gibberellin levels may promote bulblet formation in scale cuttings [5,35]. However, transcriptome analysis suggests that GAs inhibit bulblet formation in the Oriental Lily “Siberia” and may play antagonistic roles with cytokinins in the bulbil formation of *L. lancifolium* [2,36]. Additionally, the mechanisms by which GAs regulate bulblet or bulbil development remain unclear.

In this study, we investigated the effects of GAs on bulbil development. Transcriptome analysis of bulbil development and GA_3_/Paclobutrazol (PAC, gibberellin synthesis inhibitors) treatments identified key genes in the GA biosynthesis pathway (e.g., *LlGA20ox2*) and in the GA signaling pathway (e.g., *LlGID1C* and *LlCIGR2*). The functions of these three genes were validated using a VIGS (virus-induced gene silencing) system targeting bulbil development. Additionally, pathways like pentose and glucuronate interconversions and galactose metabolism may be involved in GA-regulated bud development. This study provides a foundation for further research into the molecular mechanisms of lily bulbil development.

## 2. Results

### 2.1. Gene Expression Patterns in Gibberellin Biosynthesis, Metabolism, and Signaling Pathways During Bulbil Development

To understand the bulbil development process, we divided it into three stages based on prior experience: the white dot stage (S1), green ball stage (S2), and mature stage (S3) (Figure 1A). In stage S1, the AXM visibly raises in the leaf axils. Between S1 and S2, the AXM differentiates into scale primordia, and the bulbil enlarges. By stage S3, the bulbil turns dark purple–black.

We previously performed RNA-seq on bulbils from stages S1 to S3 [7]. Transcriptome analysis revealed several significantly differentially expressed genes (DEGs) in the GA biosynthesis and signaling pathways. The early GA biosynthesis genes, *LlCPS* (*COPALUL DIPHOSPHATE SYNTHASE*) and *LlKS* (*ENT-KAURENE SYNTHASE*), were predominantly expressed in S1 and S2, particularly during S2. In contrast, *LlKAO* (*ENT-KAURENOIC ACID OXIDASE*) was highly expressed during the later stages of bulbil development, particularly in S2 and S3 (Figure 1B). *LlGA20oxs*, key enzyme genes in the GA biosynthesis pathway, were highly expressed in S2 or S3 in over 40% of cases. Only two *LlGA20ox* genes showed high expression during S1, suggesting that *LlGA20oxs* primarily function in the S2 or S3 stage of bulbil development. Several GA3ox family members, which catalyze the final step in bioactive GA synthesis, were predominantly expressed during S1 or S2. The expression pattern of the GA catabolic enzyme genes, *LlGA2oxs*, was similar to that of *LlGA20oxs*, indicating that the GA biosynthesis and metabolism pathways are more active during S2 and S3 compared to S1.

In the GA signaling pathway, genes that negatively regulate DELLA proteins, such as *LlGID1C* (*GA INSENSITIVE DWARF 1C*) and *LlGAMYB* (*GA-REGULATED MYB*), exhibited expression patterns similar to *LlGA20ox*, with high levels primarily in stages S2 and S3 compared to S1 (Figure 1C). Conversely, DELLA expression was predominantly low during stage S3. Approximately 65% of GA-responsive genes, including *LlCIGR* (*CHITIN-INDUCIBLE GIBBERELLIN-RESPONSIVE*) and members of the *GASA* (*GIBBERELLIC ACID STIMULATED ARABIDOPSIS*) family, were highly expressed during S2 or S3. Notably, the *LlGASA2* and *LlGASA4* levels significantly increased during S2 and S3 compared to S1, suggesting potential roles in bulbil development. Taken together, these results indicate that the gibberellin signaling pathway likely plays a crucial regulatory role in the middle and later stages of bulbil development.

### 2.2. Gibberellin Promotes Bulbil Elongation

To investigate the effects of gibberellin on bulbil development, *L. lancifolium* plants were treated with GA_3_ and the GA biosynthesis inhibitor PAC (Figure 2A). The results showed that GA_3_ treatment significantly increased the height of the *L. lancifolium* plants, whereas PAC treatment reduced the plant height. PAC treatment also significantly reduced the internode length and leaf number, while increasing the stem diameter (Figure 2B–D). An increase in the GA content within the leaf axils corresponded with a decrease in the number of bulbils. However, reducing the GA content had no significant impact on the bulbil count (Figure 2E,F). Morphological analysis revealed that GA_3_ treatment significantly increased the bulbil length, width, and weight, while PAC treatment significantly increased the bulbil width and scale thickness but had no significant effect on the bulbil length and weight (Figure 2G–I).

When further investigating GA’s effects on bulbil development, histological observations revealed that GA_3_ treatment increased the cell length at the distal end of the bulbil scales by at least 40% during the S2 and S3 stages compared to the control (Figure 2J–M). In contrast, PAC primarily inhibited cell elongation at the distal end of the bulbil scales during the S2 stage. This finding further indicates that gibberellin synthesis is most active in the S2 stage and promotes bulbil elongation by enhancing the cell length.

Since bulbils function as aerial bulblets of *L. lancifolium*, with a developmental process similar to underground bulblets, we also examined the effects of gibberellin on bulblet development (Appendix A). The results indicated that GA_3_ treatment significantly increased the bulblet length and weight but did not significantly affect the bulblet width. Conversely, PAC treatment significantly increased the bulblet width without significantly affecting the bulblet length or weight (Appendix A). The effects of gibberellin and PAC on bulblet development were consistent with their effects on bulbil development, further suggesting a similar role for gibberellin in both processes.

### 2.3. LlGA20ox2 Promotes the Development of Bulbils

To examine the effects of GA_3_ and PAC on genes involved in gibberellin biosynthesis and metabolism, we analyzed the gene expression patterns during bulbil development to identify those with significant differential expression in response to these treatments (Appendix A). Within the gibberellin biosynthesis pathway, we identified *LlGA20ox2* (16492.73011) as a differentially expressed gene among the GA20ox enzyme genes. Its Log_2_(S2/S1) and Log_2_(S3/S1) values were both approximately 1.5, and *LlGA20ox2* strongly responded to the GA_3_ and PAC treatments, indicating its sensitivity to the gibberellin content (Figure 3A,B).

Given the similarity between bulbils and bulblets, we silenced *LlGA20ox2* in the bulblets using TRV2-mediated VIGS (Figure 3C). With the reduction in *LlGA20ox2* expression (Appendix A), the development of bulblets was inhibited. Compared to the TRV2 control, the *LlGA20ox2*-TRV2 lines exhibited reduced bulblet length and width (Figure 3E,F). Furthermore, the cell length at the distal end of the outermost scales in the *LlGA20ox2*-TRV2 bulblets was 38% shorter than in the TRV2 control (Figure 3D,G), indicating that *LlGA20ox2* positively regulates bulblet development by promoting cell elongation.

We also overexpressed *LlGA20ox2* in the bulblets (Figure 4A). Overexpression of *LlGA20ox2* enhanced bulblet development by increasing the cell length, resulting in a 25% increase in the bulb length compared to the control (Figure 4C,D,F). However, *LlGA20ox2* overexpression did not significantly affect the bulblet width (Figure 4E).

To further investigate *LlGA20ox2*’s role in bulbil development, we cloned its promoter and analyzed its *cis*-acting elements. The results indicated that *LlGA20ox2* may respond to various hormones, including GA, ABA, and MeJA. Additionally, the *LlGA20ox2* promoter contained conserved binding sites for key gene families such as MADS-box, MYB, and MYC, which are crucial to plant development (Figure 5A). We also identified LlAGL2 as a potential upstream regulator of *LlGA20ox2*, providing a foundation for future studies of *LlGA20ox2*’s role in bulbil development (Figure 5B).

### 2.4. LlGID1C and LlCIGR2 Positively Regulate Bulblet Development

To investigate the effects of the GA_3_ and PAC treatments on gibberellin signaling pathway-related genes, we analyzed their expression and identified several genes with differential expression during bulbil development in response to the GA_3_ and PAC treatments (Appendix A). We specifically identified *LlGID1C* and *LlCIGR2* among the *GID1* and *CIGR* genes in *L. lancifolium*. The *LlGID1C* expression was significantly higher in the S2 and S3 stages compared to S1. Both *LlGID1C* and *LlCIGR2* showed significant responsiveness to the GA_3_ and PAC treatments, indicating their sensitivity to changes in the gibberellin levels (Figure 6A–D).

To further explore the roles of *LlGID1C* and *LlCIGR2* in bulbil development, we silenced these genes in bulblets. The *LlGID1C*-TRV2 and *LlCIGR2*-TRV2 lines both showed reduced bulblet length and width compared to the TRV2 control, confirming their positive regulatory roles in bulblet development (Figure 6E–G).

### 2.5. Multiple Metabolic Pathways Mediate Gibberellin-Regulated Bulbil Development

To explore the mechanisms by which gibberellins promote bulbil development, we performed transcriptomic analyses on the control, GA_3_-treated, and PAC-treated groups. The volcano plots showed that compared to the control group, 1993 genes were upregulated and 4224 genes were downregulated in the GA_3_ group. In the comparison between the control and PAC groups, 3110 genes were upregulated and 2684 genes were downregulated. In the GA_3_ vs. PAC comparison, 4102 genes were upregulated, while 2300 genes were downregulated (Figure 7A). Of these DEGs, 258 genes were differentially expressed across all the treatment groups (Figure 7B). GO and KEGG enrichment analyses of these 258 DEGs revealed significant enrichment of terms related to carbohydrate metabolism (Figure 7C,D).

The development of lily bulbils is closely associated with cell elongation and expansion, making carbohydrate metabolism, particularly that related to cell wall construction, crucial. We analyzed the FPKMs of DEGs involved in the glucuronate pathway, pentose phosphate pathway, as well as galactose metabolism. The pentose and glucuronate interconversion pathways include pectin esterase, pectate lyase, 6-phosphogluconolactonase, fructose-bisphosphate aldolase, pyrophosphate-fructose 6-phosphate 1-phosphotransferase, UDP-glucose 6-dehydrogenase, and xylose isomerase. Most of these DEGs are related to pectin esterase, showing high expression in the S2 or S3 stage and the lowest in S1. These genes all respond to GA_3_ treatment, suggesting that pectin esterase may play a crucial role in cell wall construction during GA-regulated bulbil development (Figure 8A,B).

The galactose metabolism pathway involves galactinol synthase, galactinol-sucrose galactosyltransferase, beta-fructofuranosidase, aldose 1-epimerase, hexokinase, and alpha-galactosidase. All these related genes showed significant responsiveness to GA. Except for aldose 1-epimerase, hexokinase, alpha-galactosidase, and galactinol-sucrose galactosyltransferase (cluster 16492.47772), the other DEGs show high expression during the S2 or S3 stage, potentially playing a positive regulatory role in bulbil development (Figure 8C,D).

## 3. Discussion

### 3.1. Identification of Key Stages in Bulbil Development

In this research, the development of bulbils was categorized into two distinct phases: the progression from axillary bud initiation to the green bulbil stage, and from there, through scale expansion, to the mature bulbil stage (Figure 1A) [7]. Prior studies indicate that the stages of development can differ between species and organs. For instance, in *Lycoris*, the propagation process was divided into four stages based on detailed observations: the competence stage, axillary bud initiation and elongation, bulblet formation, and bulblet growth and expansion [37,38]. In *Lilium davidii* var. *unicolor*, the formation of bulblets generally follows three steps: the appearance of rudimentary bulblets, bulblet formation, and subsequent bulblet development [6]. In this study, the classification of the bulbil (or bulblet) stages aligns with previous findings, encompassing the initiation stage, morphological changes (S1–S2), and the expansion stage (S2–S3). Identifying these crucial stages helps enhance our understanding of the regulatory mechanisms that govern bulbil development.

### 3.2. The Effects of Gibberellins on AXB Development

Shoot branching is controlled by a complex network of hormones, including auxin, cytokinin (CK), and strigolactone (SL). GA is often regarded as a branch inhibitor since GA biosynthesis and perception mutants in Arabidopsis, as well as GA-deficient transgenic plants in various species, exhibit increased branching [21,29]. However, GA has also been shown to promote lateral branch elongation in perennial strawberries, the woody plant *Jatropha curcas*, and hybrid poplar [32,33,39]. GA and CK synergistically promote lateral bud outgrowth, both negatively regulating *BRC1* (*BRANCHED 1*), a key inhibitor of lateral branch elongation [40]. Consequently, the role of GA in shoot branching remains unclear. Katyayini et al. [41] provided a possible explanation for GA’s role in lateral branch formation in hybrid poplar. Different GAs might function at distinct stages of lateral branch development. GA_3/6_ maintains high *GA2ox* expression and low GA_4_ levels in quiescent AXBs, while activation and outgrowth require elevated GA_1/4_ signaling, achieved through a rapid reduction of GA deactivation followed by subsequent GA biosynthesis. Since bulbils are a type of AXB, our results indicate that GA_3_ treatment promotes both bulbil and bulblet development (Figure 2 and Appendix A). GA_3_ likely enhances the bulbil length primarily by promoting cell elongation. However, further research is required to identify which specific gibberellin plays a major role during natural bulbil development.

Although bulbils are a type of AXB, their development mainly involves scale elongation and thickening, which differs from the elongation process in lateral branches [8]. GAs are widely recognized for regulating internode elongation. In perennial strawberries, AXB elongation (stolon development) is inhibited in *FveGA20ox4* mutants, but GA supplementation restores this phenotype [33]. In this study, we identified the gene *LlGA20ox2*, which is significantly upregulated during bulbil development and highly responsive to GA_3_ and PAC treatments in the S2 stage (Figure 3A,B and Appendix A). Silencing *LlGA20ox2* in bulblets reduced the cell length and inhibited their development, whereas overexpression of *LlGA20ox2* enhanced both (Figure 3C–G). This finding suggests that *LlGA20ox2* regulates bulblet development by modulating the scale cell length. In addition to elongation, gibberellins regulate the organ size, including the leaf size, by controlling cell division [42]. Our research demonstrates that silencing *LlGA20ox2*, *LlGID1C*, and *LlCIGR2* not only reduced the bulblet length but also decreased the bulblet width, ultimately hindering bulblet development (Figure 3F and Figure 6G). Increasing the cell length alone is not necessarily beneficial for bulb growth and storage; thus, further investigation is required to determine whether gibberellins influence bulbil development by affecting scale cell division.

### 3.3. Gibberellins Promote Bulbil Development by Affecting Carbohydrate Metabolism

Bulbil development is closely tied to sugar conversion and starch accumulation. During the development of bulblets, the levels of sucrose and starch increase [6]. The stage-specific upregulation of genes encoding enzymes involved in sucrose metabolism (Susy), UDP-glucose pyrophosphorylase (UGPase), and starch synthesis (starch synthase [SS], granule-bound starch synthase [GBSS], ADP-glucose pyrophosphorylase [AGPase]) indicates their critical roles in bulbil initiation and development [13]. During bulbil development, the increase in the soluble sugar content promotes the expression of Cyclin D (CycD) genes, accelerating cell division [43]. We found that GA_3_ treatment not only affects the cell length (Figure 2H–K) but also increases the weight of bulbils or bulblets (Figure 2G and Appendix A), potentially related to starch accumulation in these structures. Additionally, bulbil cell elongation and expansion require an energy supply. Transcriptomic analysis showed that DEGs responding to GA_3_ and PAC treatments are mainly enriched in carbohydrate metabolism pathways (Figure 7C,D). Therefore, by analyzing genes related to cell wall construction, we identified candidate genes involved in the glucuronate pathway, pentose phosphate pathway and galactose metabolism pathway as potential targets for understanding gibberellin-regulated bulbil development (Figure 8).

In conclusion, gibberellins primarily promote bulbil scale development by affecting the cell length, with the metabolic and signaling processes playing crucial roles in bulbil development (Figure 9).

## 4. Materials and Methods

### 4.1. Plant Materials and Treatments

Disease-free, dormant bulbs of *L. lancifolium* (3.5–5.0 cm in diameter) were collected from the National Lily Germplasm Bank at the Beijing Academy of Agricultural and Forestry Sciences (BAAFS). All the bulbs were soaked in a 1:500 diluted carbendazim solution for 30 min, rinsed with water, and planted in pots in BAAFS greenhouses. The planting conditions were 23 ± 2 °C, 40–60% relative humidity, and a 16/8 h light–dark cycle.

For the exogenous hormone treatment, GA_3_ and Paclobutrazol (PAC) were dissolved in DMSO (dimethyl sulfoxide) to prepare stock solutions. A 100 mg/L working solution was applied every four days from the onset of bulbil development until flowering (60 days post-planting). The DMSO-only treatment served as the control group. Samples from all three treatments were collected at the S2 stage of the control group, with each treatment including 10 independent plants. The bulbil samples were immediately frozen in liquid nitrogen and stored at −80 °C for RNA-seq and RT-qPCR (quantitative reverse transcription polymerase chain reaction) analysis.

### 4.2. RNA-Seq Library Construction, Sequencing, and Analysis of DEGs

Fifteen bulbil samples from various stages and treatments were collected for the RNA-seq analysis. Three biological replicates were included in the analysis. The total RNA from each sample was extracted using the RNAprep Pure Plant Plus Kit (DP441, Tiangen, Beijing, China) according to the manufacturer’ s instructions. The RNA quality was assessed with a NanoDrop 2000 Spectrophotometer (Implen, Wilmington, CA, USA), and the RNA Integrity Number (RIN) was measured using the Agilent 2100 Bioanalyzer system (Agilent Technologies, Santa Clara, CA, USA). RNA samples were included only if they met the following criteria: RIN ≥ 8.0, A260/280 ratio of 1.8 to 2.1, and a 25S:18S ratio of 1.7 to 2.0. Reverse transcription was conducted using the HiScript Ⅲ Super Mix for qPCR (+gDNA Wiper) RT Kit (Vazyme, Nanjing, China), synthesizing cDNA single strands. Fifteen cDNA libraries were prepared using the NEBNext^®^ UltraTMRNA Library Prep Kit for Illumina^®^ (NEB, Ipswich, MA, USA) according to the manufacturer’ s protocol. Moreover, 150 bp paired-end reads were generated on the Illumina NovaSeq platform (Novogene, Beijing, China).

Clean reads were obtained by removing adapter-containing sequences from the raw reads. De novo assembly of the clean dataset was performed using the Trinity program. All the unigene sequences were aligned to the Nt nucleotide database using Blastn, and to the following seven databases using Blastx: Nucleotide Sequence Database (Nt) (https://www.ncbi.nlm.nih.gov/nuccore, accessed on 18 February 2021), Non-Redundant Protein Database (Nr) (https://www.ncbi.nlm.nih.gov/refseq/about/nonredundantproteins/, accessed on 18 February 2021), Swiss-Prot protein (http://www.ebi.ac.uk/uniprot, accessed on 18 February 2021), Protein family (Pfam) (http://pfam.sanger.ac.uk, accessed on 18 February 2021), Gene Ontology (GO) (https://www.geneontology.org/, accessed on 18 February 2021), euKaryotic Orthologous Groups (KOG) (http://www.ncbi.nlm.nih.gov/COG, accessed on 18 February 2021), and KEGG orthology (KO) (https://www.kegg.jp/kegg/ko.html, accessed on 18 February 2021). A clustering heatmap analysis was conducted using BMKCloud, a free online platform for data analysis (https://www.biocloud.net/, accessed on 18 February 2021). Each unigene was functionally annotated based on the protein with the highest sequence similarity. The BLAST2GO program (http://www.blast2go.com/b2ghome, accessed on 18 February 2021) was used to retrieve the GO annotations for uniquely assembled unigenes.

The gene expression levels were quantified using the FPKM (fragments per kilobase of transcript per million mapped reads). Differentially expressed genes (DEGs) were identified using the DESeq2 R package (v1.20.0) with a *p*-value < 0.01 and an absolute fold change > 1. The functions of the DEGs were analyzed in GO (http://geneontology.org/, accessed on 20 March 2023) and KEGG (https://www.kegg.jp/, accessed on 20 March 2023) by using KOBAS software 3.0.

### 4.3. RT-qPCR

Specific primers for the RT-qPCR were designed using SnapGene software v5.0.5 (Appendix A). As in previous studies [44], the F-box family protein (FP) was amplified as the reference gene for the RT-qPCR validation. Each RT-qPCR analysis included at least three biological replicates. The experiment was performed with TB Green^®^ Premix ExTaq™ II (TaKaRa) in the Bio-Rad CFX96TM Real-Time System (Hercules, CA, USA). Each reaction involved a 30 s incubation at 95 °C, followed by 5 s at 95 °C, 15 s at 55 °C, 30 s at 72 °C, and a final 5 s step at 95 °C for 39 cycles. The relative expression levels of the target genes were calculated using the 2^−ΔΔCt^ method.

### 4.4. VIGS in Bulblets

Specific fragments of the target genes (*LlGA20ox2*, *LlGID1C*, *LlCIGR2*) were individually cloned into the pTRV2 vector. pTRV1, pTRV2, and the pTRV2-specific fragments were transformed into the Agrobacterium strain EHA105. Th target genes were silenced using VIGS in the bulblets. The middle scales of disease-free bulblets (circumference: 5 cm) were selected and cultured on water agar (7 g/L) in complete darkness for 30 days, producing bulblets (diameter: 3–4 mm) at the base. Scales with a similar size and bulblet morphology were selected and infiltrated with transformed EHA105 bacterial solution. The scales were subjected to vacuum infiltration (0.07 MPa, 5 min, 2 replicates) and incubated in complete darkness at 22 ± 2 °C. The relative change in the bulblet size was determined as the difference between the diameter at day 14 and day 0. Sixty independent scales from the TRV2 or silenced groups were measured. The bulblets were harvested for further analysis. The primers are listed in Appendix A. Student’s t-tests determined significance (* *p* < 0.05; ** *p* < 0.01).

### 4.5. Transient Gene Overexpression in Bulblets

The *LlGA20ox2* open reading frame (ORF) was cloned into the 35S: eGFP/pCAMBIA2300 vector. The constructed and empty vectors were transformed into the Agrobacterium strain EHA105 for transient gene overexpression in the scales. Bacteria were cultured to an OD_600_ of 1.0 and resuspended in buffer (10 mM MgCl_2_, 10 mM MES, pH 5.8, 200 μM acetosyringone). Disease-free middle-layer scales (circumference: 5 cm) were selected and infiltrated with the resuspension buffer. Vacuum infiltration was performed (0.07 MPa, 5 min, 2 replicates), and the scales were incubated on water agar (7 g/L) in complete darkness at 23 ± 2 °C. Infiltration was repeated every 4 days. The relative size change was calculated as the difference between the bulblet diameter at day 14 and day 0. Forty-eight independent scales from the TRV2 or silenced groups were analyzed. The bulblets were harvested for further analysis. The primers are listed in Appendix A. Student’s *t*-tests determined significance (* *p* < 0.05; ** *p* < 0.01).

### 4.6. Measurement of GA

Bulbil samples (0.2 g) from the control, GA_3_, and PAC treatments, with three biological replicates per group, were collected. In low light and on ice, 2 mL of 80% cold methanol extraction solution (containing 1 mM 2,6-di-tert-butyl-4-methylphenol) was added to each sample. The samples were ground into a homogenate using a mortar, then transferred to centrifuge tubes. The mortar was rinsed three times with 3 mL of extraction solution. The samples were extracted at 4 °C for 4 h with occasional shaking. The homogenates were centrifuged at 8000 r/min for 5 min at 4 °C, and the supernatants were collected. The residues were re-extracted with 3 mL of extraction solution at 4 °C for 1 h. After centrifugation, the supernatants were pooled and adjusted to a final volume of 10 mL. The combined solution was passed through a C18 column, dried under nitrogen gas, and dissolved in a sample dilution solution. The prepared samples were then sent to the School of Biology, China Agricultural University, for further analysis.

### 4.7. Histological Analysis

The samples, including bulbils at different stages, bulbils from different treatment groups, and bulblets from the *LlGA20ox2*-TRV2 and *LlGA20ox2*-OE lines, were fixed at 4 °C for 48 h in FAA fixative (formaldehyde, glacial acetic acid, and 70% ethanol at 1:1:18). The samples were dehydrated in graded ethanol (70%, 85%, 95%, and 100%), then treated with ethanol, benzene, and xylene for tissue clarification. Finally, they were soaked in paraffin wax at 65 °C (melting point 58–60 °C) for 3 d. Sections 10 µm thick were cut using a rotary microtome.

### 4.8. Statistical Analysis

Data were statistically analyzed using one-way ANOVA with the post hoc Tukey’s honestly significant difference (HSD) (*p* < 0.05) and Student’s *t*-test. GraphPad Prism (version 9) software was used for the analyses.

## 5. Conclusions

In this study, exogenous gibberellin (GA_3_) treatment enhance lily bulbil development by inducing cell elongation. The functional validation system confirmed that the key gibberellin synthesis gene *LlGA20ox2* and the signaling genes *LlGID1C* and *LlCIGR2* drive bulbil growth, reinforcing gibberellin’s role in this process. Meanwhile, multiple metabolic pathways, particularly the glucuronate pathway, pentose phosphate pathway and the galactose metabolism, were identified as mediators of gibberellin-regulated bulbil development.

## Figures and Tables

**Figure 1 plants-13-02965-f001:**
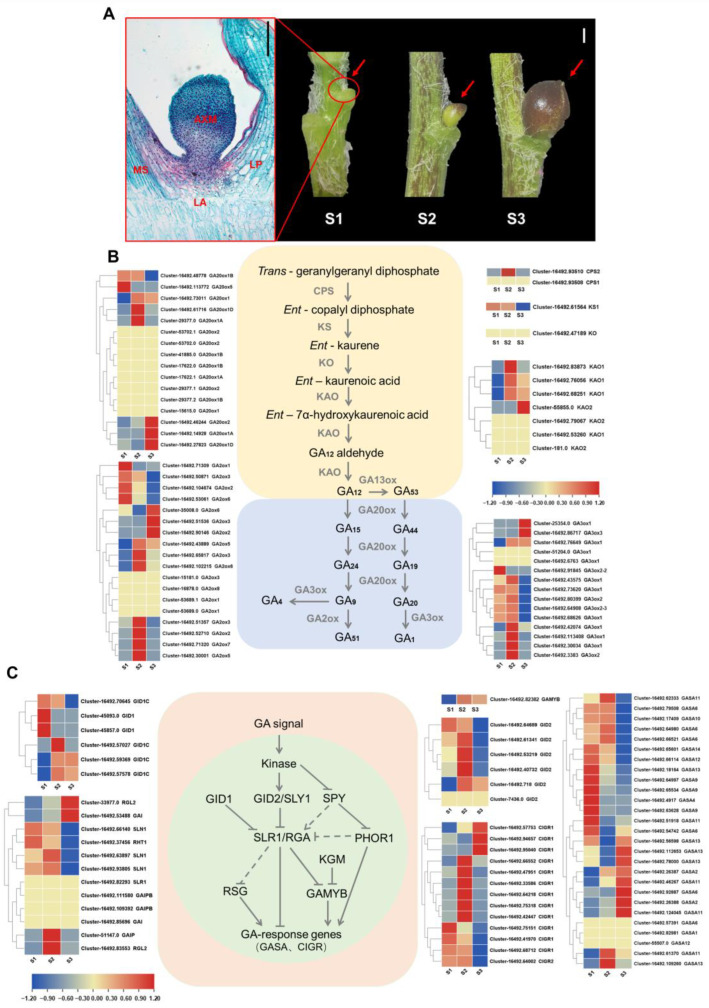
Transcriptome analyses of genes related to gibberellin synthesis, metabolism and signaling pathway during bulbil development in *Lilium lancifolium*. (**A**) Phenotypes of bulbil development in leaf axils. The red box refers to the paraffin section of the S1 stage, where the axillary meristem is visibly raised in the leaf axils. MS, main stem; LA, leaf axil; AM, axillary meristem; LP, leaf petiole. Black scale bar in red box represents 50 µm. The red arrow indicates the developing bulbil. White scale bar represents 1 mm. (**B**) Heat map showing the expression patterns of gibberellin synthesis and metabolism pathway genes during bulbil development. (**C**) Heat map showing the expression patterns of gibberellin signaling pathway genes during bulbil development. The color scale from blue to red represents the FPKM value from low to high.

**Figure 2 plants-13-02965-f002:**
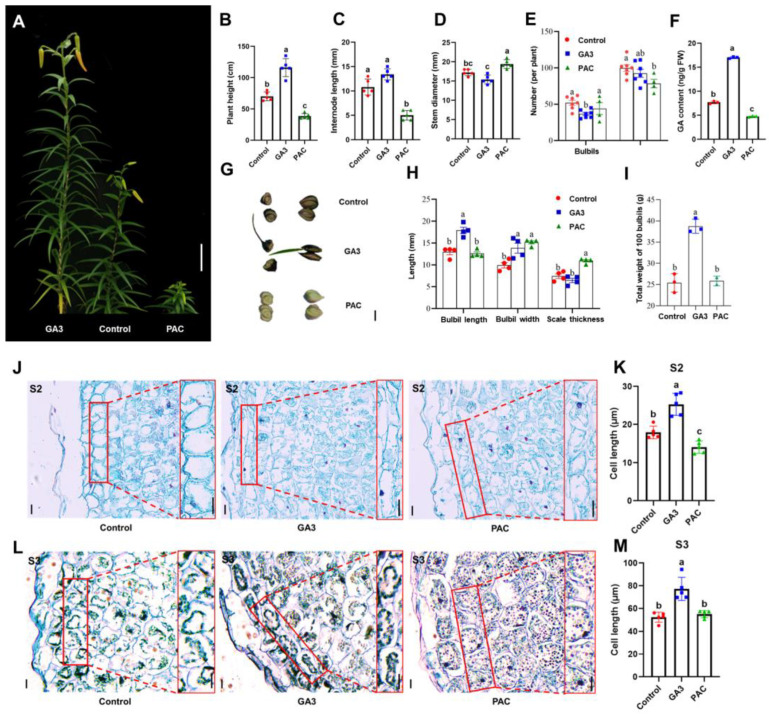
Gibberellin promotes bulbil development by modulating cell length. (**A**). The phenotypes of mock, GA_3_ and PAC treatment plants. Ten independent plants were used for each treatment. Scale bar represents 10 cm. (**B**–**D**) Plant height (**B**), internode length (**C**), and stem diameter (**D**) of plants under different treatments. (**E**) The number of bulbils and leaves in each plant. (**F**) Endogenous gibberellin contents in the leaf axils with the mock, GA_3_, and PAC treatments. At least three biological replicates were performed in panels (**B**–**F**). (**G**) The phenotypes of bulbils in mock, GA_3_ and PAC treatment plants. Scale bar represents 5 mm. (**H**) The length, width and scale thickness of bulbils in mock, GA_3_ and PAC treatment plants. (**I**) The weight of 100 bulbils in mock, GA_3_ and PAC treatment plants. Three biological replicates were performed. At least three biological replicates were performed in panels (**G**–**I**). (**J**) Histological observations of outermost scale of S2 bulbils in mock, GA_3_ and PAC treatment plants. Scale bar represents 10 µm. (**K**) Cell length at distal end of bulbil scale. (**L**) Histological observations of the outermost scale of S3 bulbils in mock, GA_3_ and PAC treatment plants. Scale bar represents 25 µm. (**M**) Cell length at distal end of bulbil scale. Three biological replicates were performed in panel and the red box is the most part of the histological observation (**J**,**L**). The lowercase letters in panels (**B**–**F**,**H**,**I**,**K**,**M**) represented significant differences calculated by an ANOVA and post hoc Tukey’ s HSD (*p* < 0.05).

**Figure 3 plants-13-02965-f003:**
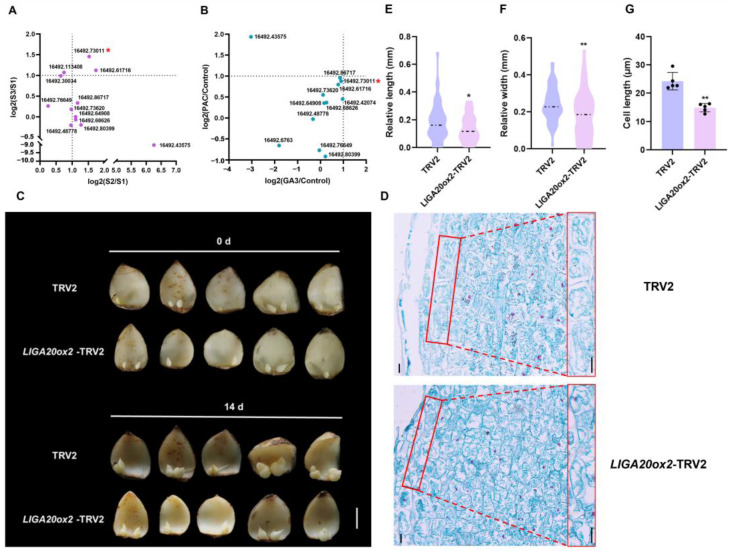
*LlGA20ox2* positively regulate lily bulbil development. (**A**) Expression analysis of GA20ox genes in bulbils at different development stages by RNAseq. *LlGA20ox2* (16492.73011) is highlighted with a red asterisk. S2: green bulbil stage. S3: mature bulbil stage. (**B**) Expression analysis of GA20ox genes in S2 bulbil at different treatment by RNAseq. *LlGA20ox2* is highlighted with a red asterisk. (**C**) Phenotypes of bulblet development on *LlGA20ox2*-TRV2 and TRV2 scales after 14 d cultivation on water agar media (7 g/L). The scale bars represent 1 cm. (**D**) Histological observations of outermost scale of bulblets in *LlGA20ox2*-TRV2 and TRV2 lines. The red box is the most part of the histological observation. The scale bars represent 10 µm. (**E**,**F**).Relative length and relative width of bulblets in the *LlGA20ox2*-TRV2 and TRV2 lines. The relative length or width was delineated as the discrepancy between the bulblet diameter at 14 d and the bulblet diameter at 0 d. Sixty independent scales in the TRV2- or *LlGA20ox2*-silenced groups were used for calculation. (**G**) Cell length at the distal end of the bulblet scale. Five biological replicates were performed. Student’ s *t*-test was used for the statistical analysis in panels (**E**–**G**) (*: *p* < 0.05; **: *p* < 0.01).

**Figure 4 plants-13-02965-f004:**
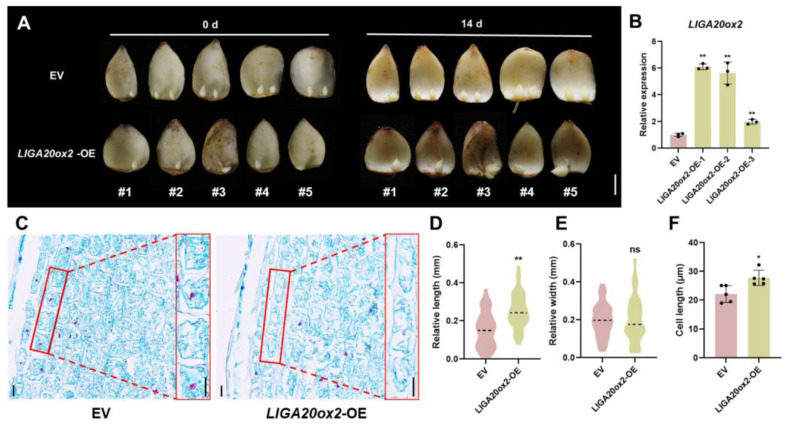
Overexpression of *LlGA20ox2* promotes bulblet development. (**A**) Phenotypes of bulblet development on the *LlGA20ox2*-OE and EV (empty vector; 35S: eGFP) scales after 14 d cultivation on water agar media (7 g/L). The scale bars represent 1 cm. (**B**) Relative expression of *LlGA20ox2* in the *LlGA20ox2*-OE and EV bulblets. (**C**) Histological observations of the outermost scale of *LlGA20ox2*-OE and EV bulblets. The red box is the most part of the histological observation. The scale bars represent 10 µm. (**D**,**E**) Relative length and relative width of the bulblets in *LlGA20ox2*-OE and EV bulblets. The relative length or width was delineated as the discrepancy between the bulblet diameter at 14 d and the bulblet diameter at 0 d. A total of 48 independent scales in the *LlGA20ox2*-OE and EV lines were used for calculation. (**F**) Cell length at the distal end of the bulblet scale in the *LlGA20ox2*-OE and EV lines. Five biological replicates were performed. Student’s *t*-test was used for statistical analysis in panels (**B**,**D**–**F**) (ns: *p* > 0.05; *: *p* < 0.05; **: *p* < 0.01).

**Figure 5 plants-13-02965-f005:**
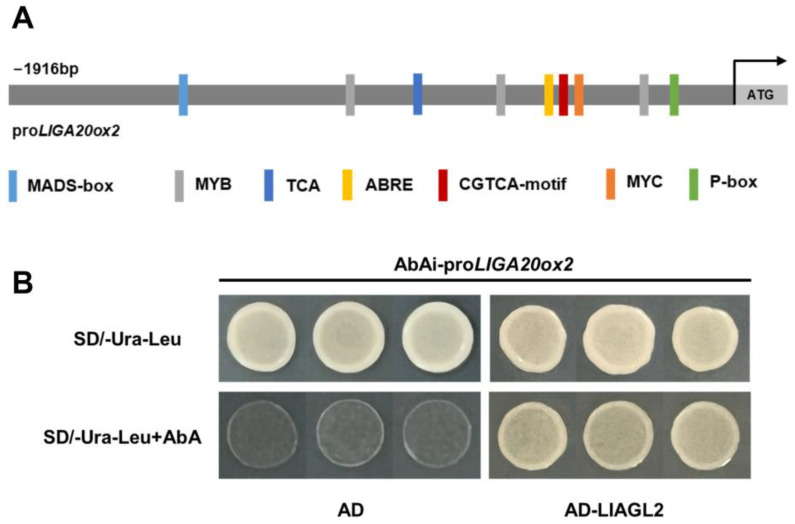
Promoter elements analysis and candidate upstream regulator of *LlGA20ox2*. (**A**) *Cis*-acting elements of *LlGA20ox* promoter. MADS-box, TATGG; MYB, CAACCA; ABRE, ACGTG, cis-acting element involved in the abscisic acid responsiveness; TCA, TCATCTTCAT; CGTCA-motif, CGTCA, *cis*-acting regulatory element involved in the MeJA-responsiveness; MYC, CATGTG; P-box, CCTTTTG, gibberellin-responsive element. (**B**) Interaction between LlAGL2 and the *LlGA20ox2* promoter (*proLlGA20ox2*) was determined in yeast one-hybrid assays by yeast growth on synthetic dropout (SD) nutrient media lacking Ura, Leu with AbA or without AbA. The same results were obtained in at least 5 independent yeast cells.

**Figure 6 plants-13-02965-f006:**
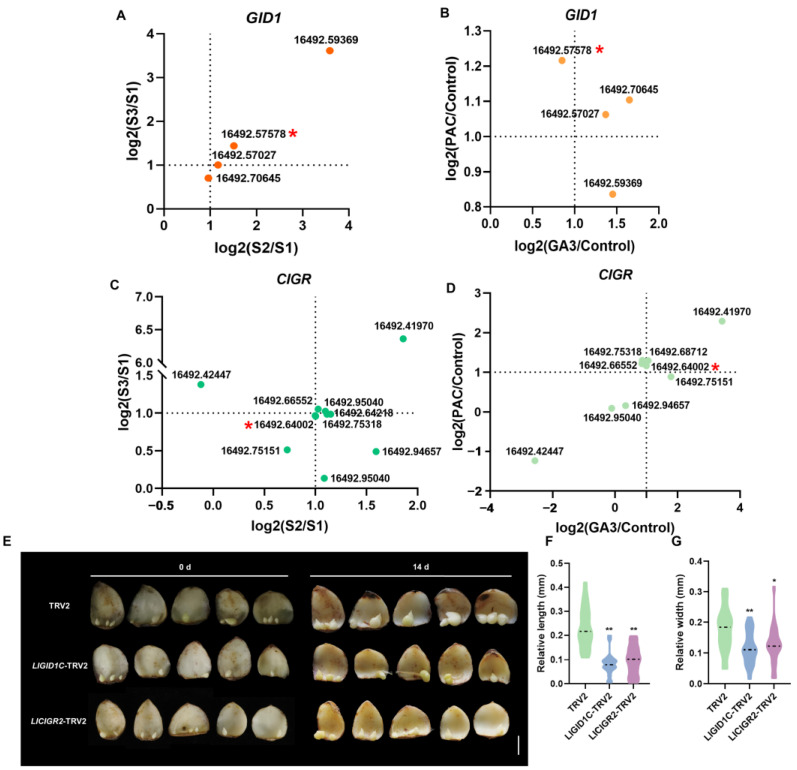
*LlGID1C* and *LlCIGR2* positively regulate bulblet development. (**A**,**C**) Expression analysis of GID1 and CIGR genes in bulbils at different development stages by RNAseq. *LlGID1C* (16492.57578) and *LlCIGR2* (16492.64002) are highlighted with a red asterisk in panel (**A**) and panel (**C**), respectively. S2: green bulbil stage. S3: mature bulbil stage. (**B**,**D**) Expression analysis of GID1 and CIGR genes in S2 bulbil at different treatment by RNAseq. *LlGID1C* and *LlCIGR2* are highlighted with a red asterisk in panel (**B**) and panel (**D**), respectively. (**E**) Phenotypes of bulblet development on *LlGID1C*-TRV2, *LlCIGR2*-TRV2 and TRV2 scales after 14 d cultivation on water agar media (7 g/L). The scale bars represent 1 cm. (**F**,**G**) Relative length and relative width of bulblets in the *LlGID1C*-TRV2, *LlCIGR2*-TRV2 and TRV2 lines. The relative length or width was delineated as the discrepancy between the bulblet diameter at 14 d and the bulblet diameter at 0 d. Sixty independent scales in TRV2 or silenced groups were used for calculation. Student’s *t*-test was used for the statistical analysis in panels (**F**,**G**) (*: *p* < 0.05; **: *p* < 0.01).

**Figure 7 plants-13-02965-f007:**
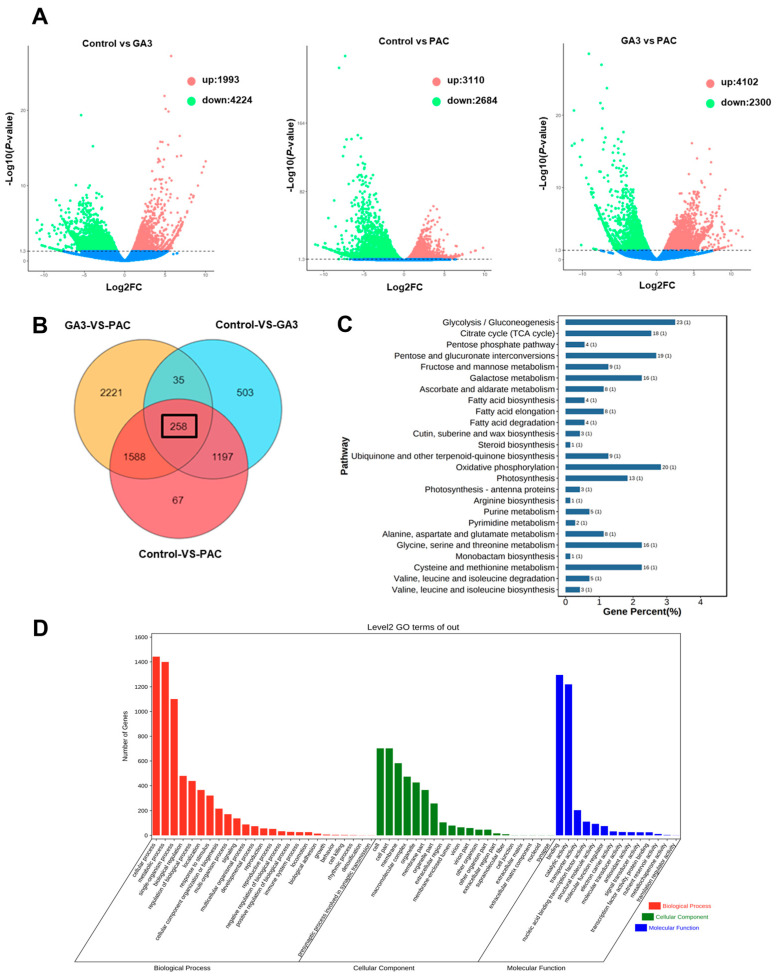
Analysis of DEGs in response to gibberellin during bulbil development. (**A**) The volcano chart shows the distribution of DEGs in the control vs. GA_3_ comparison, control vs. PAC comparison and GA_3_ vs. PAC comparison. The dotted line represents the threshold of the DEG screening criteria. The red and green dots represent up- and downregulated genes, respectively. (**B**) Venn diagram showing the number of DEGs in the control versus GA_3_, control versus PAC and GA_3_ versus PAC comparison. (**C**,**D**) Significantly enriched KEGG (**C**) and GO (**D**) pathways of 258 DEGs. The dot color means the *p*-value, and the dot size represents the gene number enriched in the corresponding pathway.

**Figure 8 plants-13-02965-f008:**
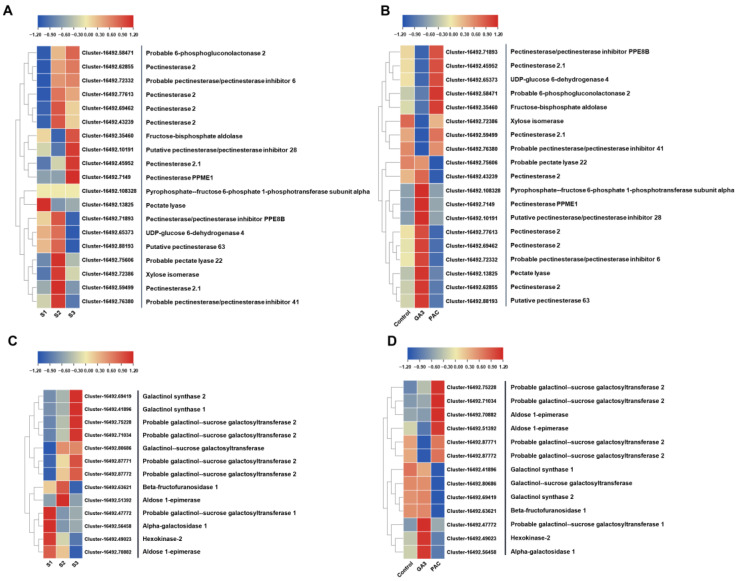
Heatmap shows the expression patterns of DEGs related to pentose and glucuronate interconversions and galactose metabolism in bulbils at different stages (**A**,**C**) during bulbil development and the mock, IAA, and NPA treatments (**B**,**D**) of the S2 bulbil. The color scale from blue to red represents the FPKM value from low to high.

**Figure 9 plants-13-02965-f009:**
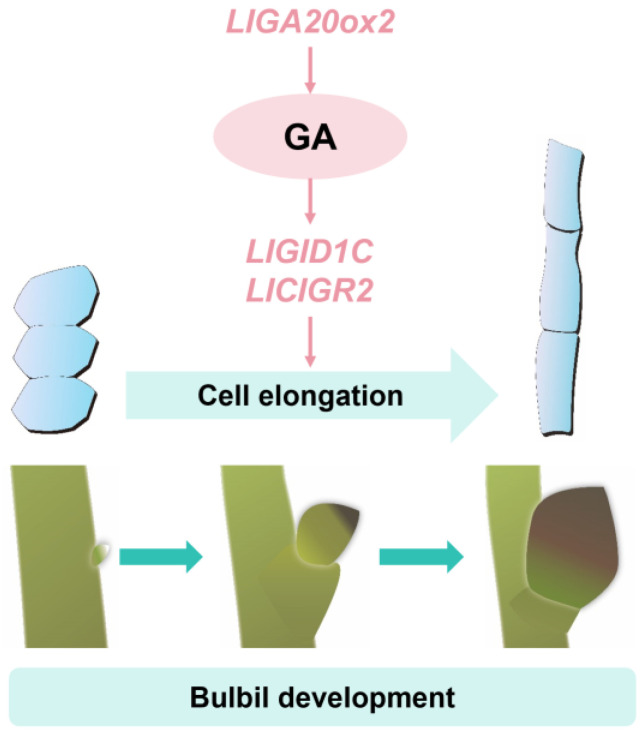
A model of gibberellin in bulbil development in *Lilium lancifolium*. The dark green arrows represent the bulbil development process in *Lilium lancifolium*.

## Data Availability

Data are contained within the article/Appendix A.

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
