# Peer review of "Histological, Transcriptomic, and Functional Analyses Reveal the Role of Gibberellin in Bulbil Development in Lilium lancifolium"

_plants, 2024, doi:10.3390/plants13212965_

Round 1
Reviewer 1 Report
Comments and Suggestions for Authors
Article Histological, transcriptomic, and functional analyses reveal the role of gibberellin in bulbil development in Lilium lancifolium by Shanshan Du, Mengdi Wang, Jiahui Liang, Wenqiang Pan, Qianzi Sang, Yanfang Ma, Mengzhu Jin, Mingfang Zhang, Xiuhai Zhang, Yunpeng Du considers the issue of bulbil enlargement in connection with gibberellin treatment.
The manuscript is of high quality and formatted according to the rules and is probably of interest for studying gene expression in lilies. Unfortunately, I cannot agree with the authors that an increase in the cell size in bulbils is good, since the water content is higher, which means the quality and protection from pathogens are lower. But the work itself is done carefully and can be useful for further work. The most important fundamental result is not obvious judging by the results, but the authors probably established the impossibility of increasing the number of cells in bulbils when using this hormone. I recommend that the authors discuss this important result in the discussion and remove the unambiguous reasoning about the benefits of increasing the bulbs by replacing them with softer and more hypothetical ones, since hydrated bulbs dry out faster and lose turgor and are more susceptible to biotic damage. However, if the authors want to prove the opposite, they only need to provide data on the ratio of dry to wet weight, but I think it will be expected and this ratio will be higher without treatments. I believe that with careful editing, the article can be published in the Plants journal.
Author Response
Response: Thanks for your wonderful suggestion. As you recommend, we have rephrased discussion section, and remove the unambiguous reasoning about the benefits of increasing the bulbs (See Line 362-364). The development process of lily bulbils is the result of cell division and expansion. In this article, we mainly focused on gibberellin through LIGA20ox2, LIGID1C and LICIGR2 genes to regulate bulbil development by promoting cell elongation at the base of bulbil scales, but this is only a small part of bulbil development. We still need to explore whether gibberellin affects the process of cell division to regulate the development of bulbils in the future (See Line 374-376).
Reviewer 2 Report
Comments and Suggestions for Authors
The present paper is devoted to the study of the role of gibberellin in the bulbil development in Lilium lancifolium plants. There are a lot of interesting data which are well described in the MS. The methods are modern and adequate. The overall impression is that the paper is solid and should be published in Plants after some minor revisions.
1. Abstract. P. 1, L. 26.
“…genes enriched in pentose…”
I think it's written incorrectly. The products of these genes are involved in described pathways. This phrase occurs in the Discussion and Conclusions. Please rephrase.
2. The data are discussed in detail in the Results section, but it might be more appropriate to discuss them in the Discussion section. But I don't insist on that.
3. Section ''3.1. Identification of key stages in bulbil development''
The section seems unfinished.
The authors wrote ''In this study, the classification of bulbil (or bulblet) stages is consistent with these earlier findings....'' but above 3 distinct stage classification were described. Please clarify and write more clearly.
Comments on the Quality of English LanguageOverall, the language is good. Only some minor changes may be made.
For instance, P. 12, L. 368-370.
“Transcriptomic data analysis revealed that DEGs induced by GA3 and PAC treatments are primarily enriched in carbohydrate metabolism pathways (Figure 7C-D).”
This is not correct in meaning, but it may be due to incorrect construction of phrases.
Author Response
Conmments 1: Abstract. P. 1, L. 26.
“…genes enriched in pentose…”
I think it's written incorrectly. The products of these genes are involved in described pathways. This phrase occurs in the Discussion and Conclusions. Please rephrase.
Response1: The description has been revised in the Abstract, Discussion, and Conclusion sections of the updated manuscript (see Lines 27, Line 294-295, Line 379-381, Line 499-501).
Comments 2: The data are discussed in detail in the Resultssection, but it might be more appropriate to discuss them in the Discussionsection. But I don't insist on that.
Response 2: Thank you for your suggestion. Since a significant portion of the Results section describes gene expression trends in the RNA-seq, we have added some discussion to enhance the clarity of the manuscript. This approach is based on several transcriptome studies (Yang et al., 2017; Xu et al., 2020), some of which even merge the Results and Discussion sections (Li et al., 2019).
Li, Y.P.; Feng, J.; Cheng, L.C.; Dai, C.; Gao, Q.; Liu, Z.C.; Kang, C.Y. Gene Expression Profiling of the Shoot Meristematic Tissues in Woodland Strawberry Fragaria vesca. Frontiers in Plant Science 2019, 10.
Xu, J.; Li, Q.; Yang, L.; Li, X.; Wang, Z.; Zhang, Y. Changes in carbohydrate metabolism and endogenous hormone regulation during bulblet initiation and development in Lycoris radiata. Bmc Plant Biol 2020, 20, 180.
Yang, P.; Xu, L.; Xu, H.; Tang, Y.; He, G.; Cao, Y.; Feng, Y.; Yuan, S.; Ming, J. Histological and transcriptomic analysis during bulbil formation in Lilium lancifolium. Front Plant Sci 2017, 8, 1508.
Comments 3:Section ''3.1. Identification of key stages in bulbil development''
The section seems unfinished.
The authors wrote ''In this study, the classification of bulbil (or bulblet) stages is consistent with these earlier findings....'' but above 3 distinct stage classification were described. Please clarify and write more clearly.
Response 3: Thank you for your suggestion. In the original version, we also described three distinct stages: after the initiation of the bulblet, the stages of morphological change and expansion followed. This description is basically consistent with other studies (Li et al. 2014; Ren et al., 2017, 2021). To enhance clarity, we have rephrased the sentence as: In this study, the classification of bulbil (or bulblet) stages aligns with previous findings, encompassing the initiation stage, morphological changes (S1-S2), and the expansion stage (S2-S3) (See Line 326-328).
Li, X.; Wang, C.; Cheng, J.; Zhang, J.; da Silva, J.; Liu, X.; Duan, X.; Li, T.; Sun, H. Transcriptome analysis of carbohydrate metabolism during bulblet formation and development in Lilium davidii var. unicolor. Bmc Plant Biol 2014, 14, 358.
Ren, Z.; Xia, Y.; Zhang, D.; Li, Y.; Wu, Y. Cytological analysis of the bulblet initiation and development in Lycoris species. Sci Hortic-Amsterdam 2017, 218, 72-79.
Ren, Z.; Xu, Y.; Lvy, X.; Zhang, D.; Gao, C.; Lin, Y.; Liu, Y.; Wu, Y.; Xia, Y. Early Sucrose Degradation and the Dominant Sucrose Cleavage Pattern Influence Lycoris sprengeri Bulblet Regeneration In Vitro. International Journal of Molecular Sciences 2021, 22, 11890.
Comments on the Quality of English Language2:
Overall, the language is good. Only some minor changes may be made.
For instance, P. 12, L. 368-370.
“Transcriptomic data analysis revealed that DEGs induced by GA3 and PAC treatments are primarily enriched in carbohydrate metabolism pathways (Figure 7C-D).”
This is not correct in meaning, but it may be due to incorrect construction of phrases.
Response: Thanks for your carefully work. The sentence has been rephrased as: Transcriptomic analysis showed that DEGs responding to GA3 and PAC treatments are mainly enriched in carbohydrate metabolism pathways (Figure 7C-D) (See Line 376-378).
Reviewer 3 Report
Comments and Suggestions for Authors
Manuscript 3228278 describes the role of gibberellins in the bulbil development of Lilium lancifolium. The story is interesting and has practical applications in the flower industry.
A couple of points to increase the readability of the manuscript as follows:
1) Figure 1B needs revision since the names of the transcripts are the same; for instance, there are 3 GA20ox1B, etc. Please correct the names of the transcripts in all figures presenting heat maps.
2) Spit Figure 2B; separate plant height from the rest. Describe what Figure 2E shows
3) Figure 4, please show the transgenic line used in pictures A, C, D, E, F
Author Response
Comments 1: Figure 1B needs revision since the names of the transcripts are the same; for instance, there are 3 GA20ox1B, etc. Please correct the names of the transcripts in all figures presenting heat maps.
Response 1: We appreciate your comments. As we understand, different transcripts may encode the same gene. For example, we identified three GA20ox1B transcripts in the transcriptome, thus, we included all three GA20ox1B transcripts in the heatmap (Figure 1B). This presentation draws on other studies (Yang et al., 2017; Xu et al., 2020; Fang et al., 2022).
Fang, S.; Yang, C.; Ali, M.; Lin, M.; Tian, S.; Zhang, L.; Chen, F.; Lin, Z. Transcriptome Analysis Reveals the Molecular Regularity Mechanism Underlying Stem Bulblet Formation in Oriental Lily 'Siberia'; Functional Characterization of the LoLOB18 Gene. Int J Mol Sci 2022, 23, 15246.
Xu, J.; Li, Q.; Yang, L.; Li, X.; Wang, Z.; Zhang, Y. Changes in carbohydrate metabolism and endogenous hormone regulation during bulblet initiation and development in Lycoris radiata. Bmc Plant Biol 2020, 20, 180.
Yang, P.; Xu, L.; Xu, H.; Tang, Y.; He, G.; Cao, Y.; Feng, Y.; Yuan, S.; Ming, J. Histological and transcriptomic analysis during bulbil formation in Lilium lancifolium. Front Plant Sci 2017, 8, 1508.
Comments 2: Spit Figure 2B; separate plant height from the rest. Describe what Figure 2E shows
Response 2: Thanks for your carefully work. We have separated plant height from the rest in Figure 2 (Figure 2B-D). And we have made a detailed description of the results of the Figure 2E (See Line 169-170).
Comments 3: Figure 4, please show the transgenic line used in pictures A, C, D, E, F
Response 3: Thanks for your carefully work. We have shown the transgenic lines in Figure 4A. However, Figure C-F is a comprehensive reflection of biological duplication, and the transgenic lines cannot be listed separately.